# Immunogenicity of an Extended Dose Interval for the Ad26.ZEBOV, MVA-BN-Filo Ebola Vaccine Regimen in Adults and Children in the Democratic Republic of the Congo

**DOI:** 10.3390/vaccines12080828

**Published:** 2024-07-23

**Authors:** Edward Man-Lik Choi, Kambale Kasonia, Hugo Kavunga-Membo, Daniel Mukadi-Bamuleka, Aboubacar Soumah, Zephyrin Mossoko, Tansy Edwards, Darius Tetsa-Tata, Rockyath Makarimi, Oumar Toure, Grace Mambula, Hannah Brindle, Anton Camacho, Nicholas E. Connor, Pierre Mukadi, Chelsea McLean, Babajide Keshinro, Auguste Gaddah, Cynthia Robinson, Kerstin Luhn, Julie Foster, Chrissy h. Roberts, John Emery Johnson, Nathalie Imbault, Daniel G. Bausch, Rebecca F. Grais, Deborah Watson-Jones, Jean Jacques Muyembe-Tamfum

**Affiliations:** 1Faculty of Infectious and Tropical Diseases, London School of Hygiene & Tropical Medicine, London WC1E 7HT, UK; kambale.kasonia@lshtm.ac.uk (K.K.); darius.tetsatata@lshtm.ac.uk (D.T.-T.); hannah.brindle@lshtm.ac.uk (H.B.); nicholas.connor@lshtm.ac.uk (N.E.C.); chrissy.roberts@lshtm.ac.uk (C.h.R.); daniel.bausch@finddx.org (D.G.B.); deborah.watson-jones@lshtm.ac.uk (D.W.-J.); 2Institut National de Recherche Biomédicale, Kinshasa P.O. Box 1192, Democratic Republic of the Congo; hugokavunga@gmail.com (H.K.-M.); drmukadi@gmail.com (D.M.-B.); zmossoko@yahoo.fr (Z.M.); kepha4@hotmail.com (P.M.); jjmuyembet@gmail.com (J.J.M.-T.); 3Epicentre, 75019 Paris, France; soumahaboubacar82@yahoo.fr (A.S.); costoreau@gmail.com (R.M.); oumar.toure@epicentre.msf.org (O.T.); grace.mambula@epicentre.msf.org (G.M.); anton.camacho@epicentre.msf.org (A.C.); rebecca.grais@epicentre.msf.org (R.F.G.); 4MRC International Statistics and Epidemiology Group, Faculty of Epidemiology and Population Health, London School of Hygiene & Tropical Medicine, London WC1E 7HT, UK; tansy.edwards@lshtm.ac.uk; 5School of Tropical Medicine and Global Health, Nagasaki University, Nagasaki 852-8523, Japan; 6Janssen Vaccines and Prevention B.V., 2333 CN Leiden, The Netherlands; cmclean@its.jnj.com (C.M.); bkeshinr@its.jnj.com (B.K.); crobin29@its.jnj.com (C.R.); kluhn@its.jnj.com (K.L.); 7Janssen Research & Development, 2340 Beerse, Belgium; agaddah@its.jnj.com; 8Médecins Sans Frontières, 75019 Paris, France; john.johnson@paris.msf.org; 9Coalition for Epidemic Preparedness Innovations, 0191 Oslo, Norway; nathalie.imbault@cepi.net; 10FIND, 1218 Geneva, Switzerland; 11Mwanza Intervention Trials Unit, National Institute for Medical Research, Mwanza P.O. Box 11936, Tanzania

**Keywords:** Ebola, outbreak, Democratic Republic of the Congo, DRC, vaccine, immunogenicity, interval, Ad26.ZEBOV, Zabdeno, MVA-BN-Filo, Mvabea

## Abstract

During the 2018–2020 Ebola virus disease outbreak in Democratic Republic of the Congo, a phase 3 trial of the Ad26.ZEBOV, MVA-BN-Filo Ebola vaccine (DRC-EB-001) commenced in Goma, with participants being offered the two-dose regimen given 56 days apart. Suspension of trial activities in 2020 due to the COVID-19 pandemic led to some participants receiving a late dose 2 outside the planned interval. Blood samples were collected from adults, adolescents, and children prior to their delayed dose 2 vaccination and 21 days after, and tested for IgG binding antibodies against Ebola virus glycoprotein using the Filovirus Animal Nonclinical Group (FANG) ELISA. Results from 133 participants showed a median two-dose interval of 9.3 months. The pre-dose 2 antibody geometric mean concentration (GMC) was 217 ELISA Units (EU)/mL (95% CI 157; 301) in adults, 378 EU/mL (281; 510) in adolescents, and 558 EU/mL (471; 661) in children. At 21 days post-dose 2, the GMC increased to 22,194 EU/mL (16,726; 29,449) in adults, 37,896 EU/mL (29,985; 47,893) in adolescents, and 34,652 EU/mL (27,906; 43,028) in children. Participants receiving a delayed dose 2 had a higher GMC at 21 days post-dose 2 than those who received a standard 56-day regimen in other African trials, but similar to those who received the regimen with an extended interval.

## 1. Introduction

From 2018 to 2020, the Democratic Republic of the Congo (DRC) experienced its tenth Ebola virus disease (EVD) epidemic, with the epicentre in North Kivu Province [1]. Despite control efforts, including ring-vaccination of EVD case contacts and contacts of contacts using the rVSV-ZEBOV-GP vaccine (Ervebo^®^, Merck & Co., Inc., Whitehouse Station, NJ, USA) [2], new cases continued to emerge, and this outbreak became the largest EVD outbreak in the DRC, and second largest on record. In May 2019, the WHO Strategic Advisory Group of Experts on Immunization (SAGE) recommended deploying a second investigational EVD vaccine to populations at some risk of exposure with provision of informed consent [3].

In November 2019, the DRC Ministry of Health, together with consortium partners, implemented a large-scale, open-label, non-randomised phase 3 trial (DRC-EB-001) that planned to vaccinate up to 500,000 healthy eligible individuals aged one year or above in North Kivu. The study began in the city of Goma and aimed to assess the vaccine effectiveness of the adenovirus serotype 26 vector-based vaccine encoding the Ebola virus (EBOV) Mayinga glycoprotein (Ad26.ZEBOV) as dose 1 and the modified Vaccinia Ankara Bavarian Nordic vector-based vaccine encoding multiple filovirus proteins (MVA-BN-Filo) as dose 2, with the two doses given at a recommended interval of 56 days (window −14/+28 days) [4,5].

In a stringent animal model, the Ad26.ZEBOV, MVA-BN-Filo regimen with a 56-day interval protected vaccinated non-human primates from lethal EBOV challenges [6]. In previous phase 1 and phase 2 trials, this regimen with a 56-day interval produced stronger binding and neutralising antibody responses than regimens using shorter intervals (14 or 28 days) [7,8,9,10]. Vaccine immunogenicity did not diminish when adult volunteers received dose 2 more than 56 days after dose 1 [8,9,11]. However, there are no data on delayed dose 2 vaccination amongst children.

The DRC-EB-001 trial enrolled and vaccinated over 20,000 participants with dose 1 within the first four months. As COVID-19 became a pandemic, the trial suspended vaccination activities in April 2020, which were not resumed until five months later. In addition, the North Kivu EVD epidemic was declared over in June 2020 [12]. While the lack of confirmed EVD cases in the study area made it impossible to evaluate vaccine effectiveness, the delayed dose 2 vaccination of over 6000 participants dictated by the COVID-19 pandemic presented an opportunity to explore immune responses to the Ad26.ZEBOV, MVA-BN-Filo regimen outside the recommended 56-day interval. After discussion and approval from the trial’s Data Safety Monitoring Board, the consortium agreed to investigate the impact of the dose 2 delay on vaccine immunogenicity in adults, adolescents, and children, and to compare the results with earlier trials that used the same two-dose regimen. The results are presented here.

## 2. Materials and Methods

### 2.1. Study Design

This study is an exploratory component of the DRC-EB-001 phase 3 trial. The main trial is registered at Clinicaltrials.gov (NCT04152486). Full study methods for the main DRC-EB-001 trial have been previously described [4]. Key safety results on the full vaccinated cohort are presented in a companion paper to this manuscript [13].

### 2.2. Study Participants

From 30 October 2020 to 16 November 2020, DRC-EB-001 participants who returned to the six study vaccination centres in the morning for a delayed dose 2 vaccination were invited to join the immunogenicity subset (Figure 1). All immunogenicity visits were completed by 7 December 2020. The target population comprised at least 50 adult participants aged 18 years or over, approximately 25 adolescents aged 12–17 years and approximately 25 children aged 4–11 years. When study activities resumed in late 2020, participants were re-consented and assessed for eligibility to join this immunogenicity substudy prior to dose 2 vaccination. Individuals were deemed eligible if they met the original inclusion and exclusion criteria in the main study protocol, did not have any contraindication to dose 2, were reachable by phone, and were willing to donate venous blood samples at two visits (Appendix A) [4]. Individuals who had received another EVD vaccine (rVSV-ZEBOV-GP) and those who missed a Filovirus Animal Nonclinical Group (FANG) ELISA result for one or more time points were excluded from immunogenicity analysis. The DRC-EB-001 trial did not require any safety laboratory assessments as a screening step for inclusion.

### 2.3. Procedures

Venous blood samples for the determination of humoral immune responses were collected prior to vaccination on the day of the dose 2 visit and 21 days post-dose 2 (window −3/+7 days). The sample volume was 5 mL per visit for participants aged six years or over, and 2.5 mL per visit for participants aged four and five years. The blood samples were processed at the Institut National de la Recherche Biomédicale laboratory in Goma. Frozen serum aliquots were sent to the Q^2^ Solutions Laboratory (San Juan Capistrano, CA, USA) for immunogenicity evaluations [14].

### 2.4. Vaccines

Ad26.ZEBOV (Zabdeno^®^, Janssen Pharmaceutica, Beerse, Belgium) is an adenovirus serotype 26 (Ad26)-vectored vaccine, encoding the EBOV Mayinga glycoprotein (GP) and given as an intramuscular injection of 0.5 mL liquid suspension, containing 5 × 10^10^ virus particles [15]. MVA-BN-Filo (Mvabea^®^, Janssen Pharmaceutica, Beerse, Belgium) is a recombinant modified Vaccinia Ankara (MVA)-vectored vaccine which is highly attenuated and does not replicate in human cells. It is multivalent, encoding the GP of EBOV (100% homologous to the EBOV GP in Ad26.ZEBOV), Sudan, and Marburg Musoke viruses, and the nucleoprotein of Taï Forest virus [16]. MVA-BN-Filo was given as an intramuscular injection of 0.5 mL liquid suspension, containing 1 × 10^8^ infectious units. Ad26.ZEBOV was manufactured by IDT Biologika GmbH, Dessau-Roßlau, Germany, and MVA-BN-Filo by Bavarian Nordic A/S, Kvistgård, Denmark, for Janssen Vaccines & Prevention B.V., Leiden, The Netherlands. Neither vaccine contains any adjuvant [4,13].

### 2.5. Immunogenicity Evaluations

Antibody responses against EBOV GP were measured by the FANG enzyme-linked immunosorbent assay (ELISA) at Q^2^ Solutions Laboratory (San Juan Capistrano, CA, USA). This assay has been validated for testing human serum [14]. The concentrations of IgG antibodies binding to recombinant EBOV GP were presented as geometric mean concentrations (GMCs) in ELISA Units (EU)/mL. The lower limit of quantification (LLOQ) for this ELISA, as determined in the assay validation, was 36.11 EU/mL. Samples with results above the upper limit of quantification (ULOQ) were diluted in assay diluent and retested. The retesting results were multiplied by the pre-dilution factor to determine a final result.

### 2.6. Data Management

Participant data were collected on a combination of paper and electronic case report forms (CRFs). Data collected on paper CRFs were double-entered into a REDCap database (version 11.0.3). Electronic CRFs completed on tablets using ODK (ODK Collect v1.16 to v1.22, ODK Aggregate v1.44 to v2.03) were encrypted before syncing to a secure server and imported into REDCap, as previously described [4]. Query resolution and quality assurance were performed on an ongoing basis by members of the data team, quality team and the internal monitoring team.

### 2.7. Statistical Analysis

The sample size calculations were based on estimating the GMC with precision with no formal statistical hypothesis testing considerations. We assumed a standard deviation of 0.55 (on log_10_-scale) of the binding antibody concentrations as measured by ELISA at 21 days post dose 2, based on the maximum observed standard deviation in the EBL3001 study plus 10% [11,17]. It was estimated that a sample size of 50 would allow estimation of a two-sided 95% confidence interval with a distance from the mean to the limits that is equal to 0.156 (on log_10_-scale).

Individuals with FANG ELISA results for both pre-dose 2 and 21 days post-dose 2 samples (i.e., paired samples) were included in the immunogenicity analysis population.

The age and sex distributions were described, along with the mean, standard deviation (SD), median, and minimum–maximum range of the time intervals between dose 1 and dose 2 in days and months (28 days). The GMC of antibodies and corresponding 95% confidence intervals (CI) were calculated per time point for all participants with paired samples within the defined age groups for children, adolescents, and adults.

Fold change (fold increase) of antibody concentrations was calculated as the ratio of the 21 days post-dose 2 to the pre-dose 2 GMC value on the original measurement scale before calculation of geometric mean fold change and corresponding 95% CIs for each age group. Differences in values at 21 days post-dose 2 by age group, conditional on pre-dose 2 (day 0) values, were tested using ANOVA, analysing data on the log base_10_ scale with the outcome as the 21 days post-dose 2 values, adjusting for pre-dose 2 (day 0) values. Graphical presentation of data included jittered boxplots to show the distribution of immunogenicity data by time and age group, and confidence interval plots to show summary data by time and age group. The LLOQ is shown as 36.11 and the ULOQ as 194,938.9. Results above the ULOQ were imputed as 194,939 EU/mL.

Because our DRC trial and other African trials of the same vaccine were conducted in different populations using different inclusion/exclusion criteria, no formal statistical testing of their comparison was performed.

### 2.8. Ethics Statement

The protocol was approved by the Ethics Committee/Institutional Review Board of the London School of Hygiene & Tropical Medicine (LSHTM EC; Ref 17471), the Médecins Sans Frontières Ethics (MSF-ERB; ID:1922), the University of Kinshasa School of Public Health (CE-ESP; ESP/CE/234/2019), and Comité National d’Ethique de la Santé (CNES; n°138/CNES/BN/PMMF2019) [4].

## 3. Results

### 3.1. Study Participants

In total, 161 individuals were screened and 150 gave consent and/or assent to participate in the subset and provided a blood sample prior to receiving their dose 2 vaccine (Figure 2). Seventeen participants were excluded from analysis, including two adults who had received rVSV-ZEBOV-GP in addition to the study vaccines (Appendix A), leaving a total of 133 individuals with paired pre- and post-dose 2 samples. The final Immunogenicity Analysis Population comprised 49 adults, 32 adolescents, and 52 children (Figure 2). All immunogenicity subset participants received their dose 2 outside the recommended interval of 56 days (window −14/+28 days) post-dose 1. The median (minimum–maximum) interval between dose 1 and dose 2 was 9.3 (8.7–12.1) months for adults, 9.5 (8.9–11.7) months for adolescents, and 9.1 (8.9–10.1) months for children. The baseline demographics of the participants included in the immunogenicity substudy are shown in Table 1.

### 3.2. Vaccine Immunogenicity

Upon administration of a delayed dose 2, the antibody GMC increased by 102-fold in adults, 100-fold in adolescents and 62-fold in children within 21 days (Figure 3, Table 2). In the absence of pre-dose 1 data, pre-dose 2 GMC in all participants served as a conservative baseline antibody level. All but one participant 132/133 (99%) had an antibody GMC that increased more than 2.5 times after receiving dose 2 and may thus be considered responders.

Amongst 133 paired samples, linear regression was used to model 21 days post-dose 2 data on the log_10_ scale, conditional on pre-dose 2 (day 0) values on the log base_10_ scale, to investigate differences in fold change (fold increase) by age group. The analysis showed post-dose 2 antibody values to be higher in adolescents compared to adults (regression coefficient: 0.175, 95% CI: 0.126, 0.338, *p* = 0.035), when accounting for pre-dose 2 values. However, there was no significant difference between children and adults (regression coefficient: 0.097, 95% CI: −0.056, 0.250, *p* = 0.214). More variability in pre-dose 2 values can be seen in the adult group (Appendix A).

The antibody GMC following the Ad26.ZEBOV, MVA-BN-Filo regimen with an extended two-dose interval observed in our study was higher than that reported in other sub-Saharan African trials [8,11,17,18,19,20]. Immunogenicity data stratified by the same age groups are available from the EBL3001 trial (NCT02509494) conducted in Sierra Leone and the EBL2002 trial (NCT02564523) conducted in Burkina Faso, Côte D’Ivoire, Kenya, and Uganda. Both EBL3001 and EBL2002 measured anti-EBOV GP IgG binding antibody GMC at 21 days after dose 2 using the same FANG ELISA at the same FDA-approved central laboratory as this study, thus permitting a direct comparison between three trials. As shown in Table 3, the post-dose 2 GMC observed in this study was at least 3.4 times higher than among EBL3001 participants who received the standard 56-day regimen in all three age groups. Similarly, the GMC observed in this study was at least 2.0 times higher than in healthy EBL2002 participants who received the standard regimen. There is no overlap between the 95% CIs around GMCs in these comparisons.

The heightened immunogenicity induced by late dose 2 administration in this study is reminiscent of the out-of-window antibody results obtained in EBL2002. In that trial, 169 healthy adult participants also received dose 2 beyond the protocol-defined window due to a temporary study pause [8]. As the two-dose interval extended from 56 to ≥280 days among these EBL2002 adult participants, the 21 days post-dose 2 antibody GMC increased from 7518 (95% CI 6468; 8740 EU/mL) to 23,897 EU/mL (95% CI 16,703; 34,188) (Figure 4). The GMC reported among the ≥280 days out-of-window EBL2002 adults (median interval 295 days), at 23,897 EU/mL (95% CI 16,703; 34,188), is comparable to the GMC observed among adults in this substudy (median interval 259 days), at 22,194 EU/mL (95% CI 16,726; 29,449). In EBL2002, extending the two-dose interval from 56 to ≥280 days also increased the pre- and post-dose 2 GMC fold change from 21-fold to 113-fold. The positive correlation between the length of the interval and the post-vaccination GMC can be illustrated by plotting the EBL2002 healthy adult data and fitting a curve using ordinary least square linear regression (Figure 4). When the adult data from our substudy are added to the plot, the immunogenicity of the delayed dose 2 in DRC-EB-001 appear consistent with the out-of-window immunogenicity in EBL2002.

## 4. Discussion

We measured the immunogenicity of the Ad26.ZEBOV, MVA-BN-Filo vaccine regimen in participants who received a delayed dose 2 and showed that extending the two-dose interval from the standard 56 days to approximately nine months enhanced antibody responses in adults, adolescents, or children. The EBOV GP binding antibody concentrations observed in our study are significantly higher than those reported in the EBL2002 and EBL3001 trials in other sub-Saharan African countries using the standard 56-day regimen. Furthermore, the immunogenicity in this study is consistent with the out-of-window immunogenicity in EBL2002, suggesting that the high antibody concentrations observed in our study are the result of delaying dose 2 administration. Our result is similar to previous findings that longer intervals between first and second dose can increase immunogenicity of vaccines against influenza A, malaria, and COVID-19 [21,22,23]. In the case of ChAdOx1.S nCoV-19, the Oxford-AstraZeneca adenovirus-vectored COVID-19 vaccine, extending the two-dose interval from less than six weeks to over 12 weeks increased binding and neutralising antibody responses as well as vaccine efficacy in protection against symptomatic infection [24,25].

This study has several limitations. Pre-dose 1 vaccination blood samples were not collected because the immunogenicity subset was implemented as an exploratory objective after all dose 1 vaccinations had concluded. It was therefore not possible to measure the vaccine responder rate using pre-dose 1 baseline GMC values, as in previous studies. The immunogenicity subset of the DRC-EB-001 trial did not include a comparator arm that received the standard 56-day regimen. Since all participants in the immunogenicity subset received their dose 2 vaccination between 8 to 12 months post-dose 1, direct comparison of the immunogenicity of the standard dosing schedule and the extended interval with a delayed dose 2 could not be measured within the same trial population. Because the immunogenicity follow-up period was only 21 days (window −3/+7 days), long-term persistence of the EBOV antibodies following a delayed dose could not be measured. To date there are still no clinical data on the vaccine efficacy of the Ad26.ZEBOV, MVA-BN-Filo regimen. Nor have the mechanistic correlates of protection been established. Nevertheless, the regimen’s clinical benefits were inferred by immunobridging from a stringent animal model, where vaccinated non-human primates (NHP) were 100% protected against EBOV challenge [6]. In these experiments, NHP survival was strongly associated with anti-EBOV GP binding antibody GMC. Of note, the GMCs achieved 21 days after Ad26.ZEBOV, MVA-BN-Filo vaccination in studies that implemented the standard 56-day regimen and this study that investigated a delayed dose 2 were at least as high as those achieved 28 days after vaccination with rVSV-ZEBOV-GP (Ervebo^®^), which demonstrated 100% protection against EVD in a phase 3 trial [18,26,27].

Taken together, results from the DRC-EB-001 trial add to the existing literature to support deployment of the Ad26.ZEBOV, MVA-BN-Filo vaccine for EVD as prophylactic vaccines for at-risk populations. During EVD outbreaks, every effort should be made to ensure that the complete regimen is delivered within the recommended 56-day interval in order to reach potentially protective levels of antibodies as soon as possible. However, a host of logistical challenges are often encountered in these difficult settings which may affect routine access to vaccination, including insecurity, lack of trained healthcare workers, population movement, and difficult maintenance of the cold chain [28]. Indeed, many of these were encountered during the DRC-EB-001 trial. The knowledge that, when necessary, delayed administration of dose 2 of the Ad26.ZEBOV, MVA-BN-Filo regimen has no negative impact on immunogenicity, gives comfort to the use of the vaccine even in the most turbulent settings. Public health and immunisation services should be encouraged to complete the two-dose schedule even if this is with an extended second dose window.

## 5. Conclusions

In conclusion, the Ebola vaccine regimen Ad26.ZEBOV, MVA-BN-Filo induced strong antibody responses in DRC-EB-001 trial participants who received dose 2 outside the recommended interval of 56 days (window −14/+28 days) after dose 1. Based on comparison with previous studies, delaying dose 2 administration may increase the level of anti-EBOV GP binding IgG antibody in children, adolescents, and adults. Nevertheless, because a person is still vulnerable to EBOV infection after the first dose of vaccine, maintaining the 56-day interval is still recommended when possible.

## Figures and Tables

**Figure 1 vaccines-12-00828-f001:**
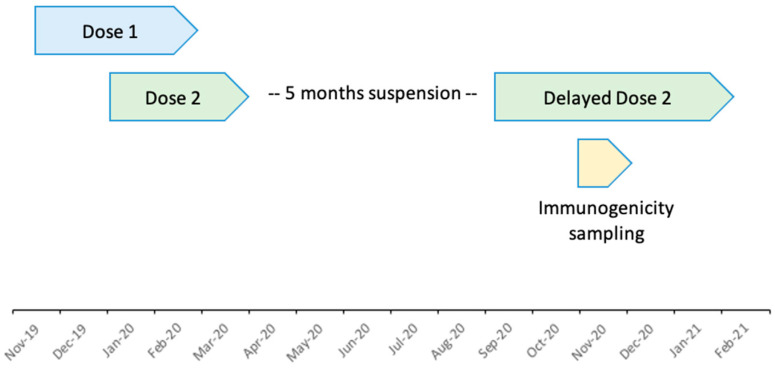
Study timeline.

**Figure 2 vaccines-12-00828-f002:**
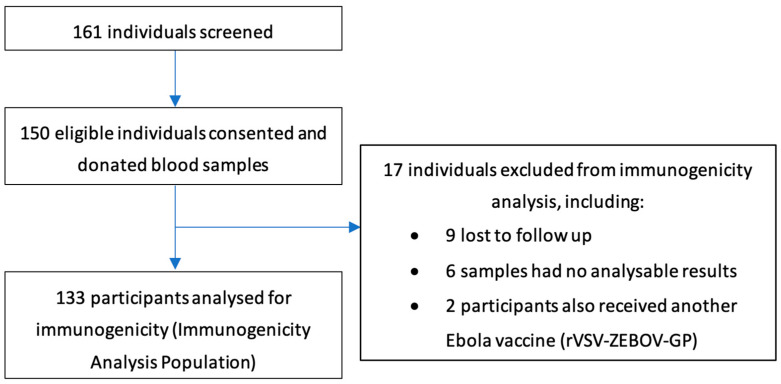
Disposition of delayed second dose immunogenicity subset participants.

**Figure 3 vaccines-12-00828-f003:**
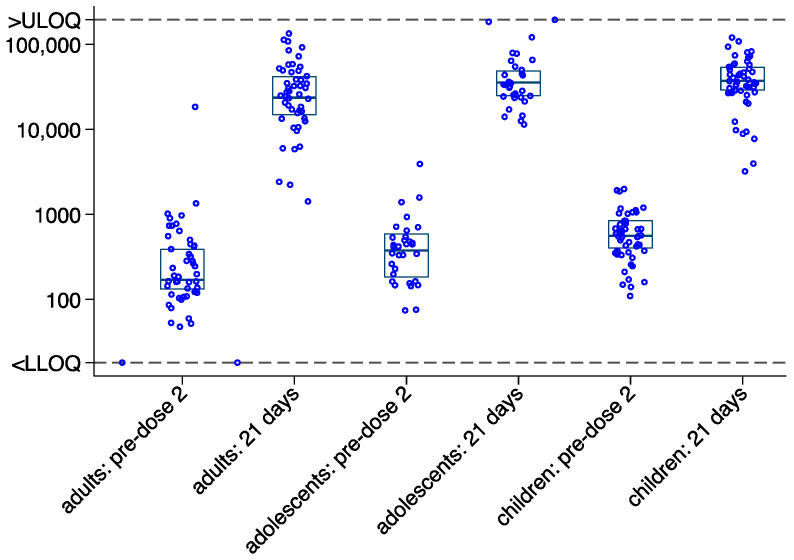
Ebola virus glycoprotein-specific binding antibody responses at pre-dose 2 and 21 days post-dose 2 in participants receiving a delayed dose 2 vaccine—distribution by age group. Antibody geometric mean concentration (GMC, ELISA Unit (EU)/mL) in adults (18 years or above), adolescents (12–17 years), and children (4–11 years). Boxes indicate median and interquartile range; EBOV (Ebola virus); LLOQ (Lower limit of quantification); ULOQ (Upper limit of quantification).

**Figure 4 vaccines-12-00828-f004:**
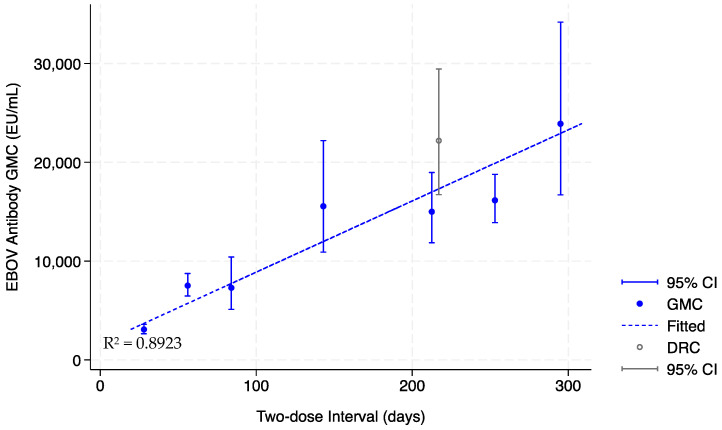
Immunogenicity of extended dose intervals for the Ad26.ZEBOV, MVA-BN-Filo vaccine regimen in this study and the EBL2002 trial. Closed circles depict immunogenicity results of healthy adult participants in the EBL2002 trial conducted in Burkina Faso, Côte D’Ivoire, Kenya, and Uganda who received dose 2 between 28 days to ≥280 days after dose 1. Open circles depict immunogenicity data collected in this DRC-EB-001 trial from immunogenicity subset adult participants in the DRC who received their dose 2 approximately 9.3 months after dose 1. A trendline (dashed line) is fitted by ordinary least square linear regression of EBL2002 data from healthy adults using STATA/SE 18.0.

**Table 1 vaccines-12-00828-t001:** Study participant demographic and baseline characteristics.

Immunogenicity Analysis Population		Adults (18+ Years)	Adolescents (12–17 Years)	Children (4–11 Years)
Participants with paired samples, n (total = 133)		49	32	52
Age (years)	Mean (SD)	31.0 (11.0)	13.5 (1.3)	6.8 (2.2)
	Median (range)	28.0 (18.0–62.0)	13.0 (12.0–17.0)	6.0 (4.0–11.0)
Sex	Female, n (%)	33 (67.3)	15 (46.9)	25 (48.1)
	Male, n (%)	16 (32.7)	17 (53.1)	27 (51.9)
Days between dose 1 and dose 2	Mean (SD)	265.4 (21.6)	269.5 (16.1)	258.0 (7.5)
	Median (range)	259.0 (244.0–338.0)	265.0 (249.0–327.0)	255.0 (248.0–283.0)
Months between dose 1 and dose 2	Mean (SD)	9.5 (0.8)	9.6 (0.6)	9.2 (0.3)
	Median (range)	9.3 (8.7–12.1)	9.5 (8.9–11.7)	9.1 (8.9–10.1)

SD, Standard deviation; range, minimum–maximum.

**Table 2 vaccines-12-00828-t002:** Ebola virus glycoprotein-specific binding antibody concentrations.

Immunogenicity Analysis Population	Adults (18+ Years)	Adolescents (12–17 Years)	Children (4–11 Years)
Participants with paired samples, n (total = 133)	49	32	52
Pre-dose 2 GMC (95% CI)	217 (157; 301)	378 (281; 510)	558 (471; 661)
21 days post-dose 2 GMC (95% CI)	22,194 (16,726; 29,449)	37,896 (29,985; 47,893)	34,652 (27,906; 43,028)
GMC fold change * between time points	102	100	62

Data are number of paired samples (n), geometric mean concentration, and 95% confidence interval (GMC, 95% CI). * Fold change = GMC at 21 days post-dose 2/GMC at pre-dose 2.

**Table 3 vaccines-12-00828-t003:** Comparison of Ebola virus glycoprotein-specific binding antibody concentrations between sub-Saharan African trials assessing the Ad26.ZEBOV, MVA-BN-Filo Ebola vaccine regimen.

Countries	Dosing Interval Median Days (Range) *	Number of Participants **	Antibody GMC at Pre-dose 2	Antibody GMC at 21 Days Post-Dose 2 ^†^	GMC Fold Change after Dose 2 ^††^	Trial Protocol [Reference]
Adults (18+ years)						
DRC	259 (244–338)	49	217 (157; 301)	22194 (16,726; 29,449)	102	DRC-EB-001
SL	56 (55–63) 54 (50–63)	42 (stage 1) 182 (stage 2)	269 (208; 347) 236 (206; 270)	4784 (3736; 6125) 3810 (3312; 4383)	18 16	EBL3001 [11]
BF/CI/KN/UG	56 (54–58)	136	361 (307; 423)	7518 (6468; 8740)	21	EBL2002 [8]
Adolescents (12–17 years)						
DRC	264 (249–327)	32	378 (281; 510)	37896 (29,985; 47,893)	100	DRC-EB-001
SL	52 (50–58)	134	314 (269; 366)	9929 (8172; 12,064)	32	EBL3001 [17]
BF/CI/KN/UG	56 (53–57)	53	562 (460; 686)	13532 (10,732; 17,061)	24	EBL2002 [19]
Children (4–11 years)						
DRC	255 (248–283)	52	558 (471; 661)	34652 (27,906; 43,028)	62	DRC-EB-001
SL	55 (51–62)	124	390 (334–456)	10212 (8419; 12,388)	26	EBL3001 [17]
BF/CI/KN/UG	56 (54–59)	53	658 (556–780)	17388 (12,973; 23,306)	26	EBL2002 [19]

* The number of days between dose 1 and 2 expressed as median (minimum–maximum range). The data presented in this table include participants in previous randomised trials who received the Ebola vaccine regimen at the standard 56-day interval but exclude those who received placebo. The dosing interval medians and ranges reported here are therefore different from those published in previous papers whereby the placebo participants were also included. ** The number of participants with immunological data available at 21 days post-dose 2. ^†^ GMC in [EU/mL] (95% CI). ^††^ Fold change = GMC at 21 days post-dose 2/GMC at pre-dose 2. GMC, Geometric mean concentration; EU, ELISA Unit; BF, Burkina Faso; CI, Côte D’Ivoire; DRC, The Democratic Republic of the Congo; KN, Kenya; SL, Sierra Leone; UG, Uganda.

## Data Availability

Following publication of the primary and exploratory outcomes as detailed in the protocol, deidentified participant data will be made available upon request via LSHTM Data Compass, the London School of Hygiene & Tropical Medicine research data repository at http://datacompass.lshtm.ac.uk (accessed on 16 August 2023). Requests with a defined analysis plan can be sent via LSHTM Data Compass. We believe that the study methods and results are clearly documented in this article. The clinical study protocol has been published with open access [4].

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
