# Peer review of "Immunogenicity of an Extended Dose Interval for the Ad26.ZEBOV, MVA-BN-Filo Ebola Vaccine Regimen in Adults and Children in the Democratic Republic of the Congo"

_vaccines, 2024, doi:10.3390/vaccines12080828_

Round 1

Reviewer 1 Report

Comments and Suggestions for Authors

This is a great paper about an important topic and reads nicely

I have several minor suggested changes

1. Line 37 reads "had higher GMC than" but I suggest this is altered to read ""had higher GMC 21 days after boosting than" to differentiate from the bleed taken on boosting, if this is what is intended

2. On line 64, there is currently an incorrect reference to reference 5. Currently, the correct reference should be 15. But the current reference 5 (about the FANG assay) is relevant to the introduction and so there probably needs to be some new text to cite current reference 5 because line 100 refers to the FANG assay without any previous mention of it. If the current reference 15 become reference 6, all references will need to be incrementally altered.

3. Does the self-reference 10 have a doi available in a preprint server? If so, please provide the doi

 4. Is it normal to cite a reference after the sentence has finished? I wold normally cite the reference at the end of a sentence and then drop in the full stop. 

5. Given that the Ebola outbreak did not cease until June 2020, is there any way to detect if any of the participants had had a subclinical infection with Ebola? Would their immunity have additional components to other viral proteins? Could this contribute to the unexpectedly higher responses in any of the participants? Perhaps you cannot retrospectively look for this? Perhaps this topic warrants some added text in the discussion? 

6. Line 183 cites the ULOD to seven significant figures. Is this level of significant figures justified? I assume this is the antilog of a number which was calculated using ELISA data which has no more than five significant figures?  Is perhaps is should be rounded to six significant figures?

Line 201, Should the Investigator's middle initial be a capital?

7. In the discussion, perhaps you could speculate as to why there was an increase in response in the delayed cohorts? Could it be that their immunity has a more mature and higher affinity antibody that binds better to solid phase ELISA plates? Could it be that the host's immune system is better primed if left for longer? Could there have been subclinical exposure during the gap? This could be addressed in association with the response to item 5 above

8. Line 384, delete the word "and" between logistical and challenges or otherwise correct the sentence.

Author Response

Comment 1

Line 37 reads "had higher GMC than" but I suggest this is altered to read ""had higher GMC 21 days after boosting than" to differentiate from the bleed taken on boosting, if this is what is intended

Response:

Thank you. We have amended the text to “had higher GMC at 21 days post-dose 2”

Comment 2

On line 64, there is currently an incorrect reference to reference 5. Currently, the correct reference should be 15. But the current reference 5 (about the FANG assay) is relevant to the introduction and so there probably needs to be some new text to cite current reference 5 because line 100 refers to the FANG assay without any previous mention of it. If the current reference 15 become reference 6, all references will need to be incrementally altered.

Response:

Thank you for spotting the error. The references have now been corrected and confirmed.

Comment 3

Does the self-reference 10 have a doi available in a preprint server? If so, please provide the doi

Response:

The manuscript for this reference is at an advanced stage of peer review by the same journal. We hope to publish the two sister papers back-to-back and provide the actual citation of this said reference in the final article.

Comment 4

Is it normal to cite a reference after the sentence has finished? I wold normally cite the reference at the end of a sentence and then drop in the full stop. 

Response:

We have updated the referencing format as suggested.

Comment 5.

Given that the Ebola outbreak did not cease until June 2020, is there any way to detect if any of the participants had had a subclinical infection with Ebola? Would their immunity have additional components to other viral proteins? Could this contribute to the unexpectedly higher responses in any of the participants? Perhaps you cannot retrospectively look for this? Perhaps this topic warrants some added text in the discussion? 

Response:

While previous infection could be assessed by testing for the presence of EBOV NP antibodies, we did not perform this and it would be very difficult to do retrospectively. Nevertheless, although we cannot completely exclude it, we think that it is unlikely that many, if any, of the trial participants had previous EVD for the following reasons: 1) As explained in the manuscript, previous clinically recognized laboratory EVD was one of the exclusion criteria., 2) Prior to this outbreak, EVD had never been reported or known to be endemic in Eastern DRC., 3) The epicentres of this outbreak were in Mabalako and Beni Health Zones, over 200 miles north of Goma and no EVD cases were detected over the duration of the trial in Goma, where our participants were recruited. 
Even if we are incorrect and there are a few participants with previous EBOV exposure, it is unlikely to be many, and thus unlikely to significantly impact the findings or our conclusions. Lastly, studies from Sierra Leone have shown that individuals with high and low concentrations of pre-existing EBOV antibodies at baseline had no significant difference in their post-vaccination antibody concentrations three weeks after completing the Ad26,MVA regimen (Unpublished results, EBOVAC1 report). 

Comment 6

Line 183 cites the ULOD to seven significant figures. Is this level of significant figures justified? I assume this is the antilog of a number which was calculated using ELISA data which has no more than five significant figures?  Is perhaps is should be rounded to six significant figures?

Response:

The ULOD has been rounded to six significant figures, as suggested.

Comment

Line 201, Should the Investigator's middle initial be a capital?

Response:

Amended

Comment 7

In the discussion, perhaps you could speculate as to why there was an increase in response in the delayed cohorts? Could it be that their immunity has a more mature and higher affinity antibody that binds better to solid phase ELISA plates? Could it be that the host's immune system is better primed if left for longer? Could there have been subclinical exposure during the gap? This could be addressed in association with the response to item 5 above

Response

The phenomenon of increasing vaccine immunogenicity and efficacy by extending the two-dose interval has previously been observed in other vaccines, including DNA/monovalent inactivated vaccine against influenza A, attenuated sporozoite vaccine against malaria, and adenovirus-vectored vaccine against SARS-CoV-2 (Ledgerwood et al 2013, Fernandez-Arias et al 2018, McQuade et al 2021). Most significantly, widening the two-dose interval of the ChAdOx1.S nCoV-19  (Oxford-AstraZeneca) SARS-CoV-2 vaccine was found to improve both immunogenicity and efficacy. In a meta-analysis of four randomised controlled trials of ChAdOx1.S nCoV-19, lengthening the two-dose interval from <6 weeks to ≥12 weeks increased vaccine protection from 55·1% [33·0–69·9] to 81·3% [95% CI 60·3–91·2], and increased binding and neutralizing antibody responses (Voysey et al 2021, Flaxman et al 2021). We do not yet know how the timing of the second dose or booster dose may affect clonal expansion/contraction of B cells and T cells, or how it may influence affinity maturation of vaccine-induced antibodies. However, the longer an individual has to wait to complete the two dose schedule, the longer they are at risk of infection. We have expanded the discussion to include some of these key points.

Comment 8

Line 384, delete the word "and" between logistical and challenges or otherwise correct the sentence.

Response:

Amended.

Reviewer 2 Report

Comments and Suggestions for Authors

The present manuscript discusses the immunogenicity in adults and children treated with the ebola vaccine. The manuscript is good but I have a few points to consider before acceptance.

1. The introduction should discuss the vaccine Ad26.ZEBOV and MVA-BN-Filo and their biochemical nature. In the current structure, it is directly focusing on clinical use. This is good. But a reader who is not a clinician and wants to know about the chemical nature, route of administration, ingredients, etc will not understand or follow the article just by the name of the vaccine.

2. I think the authors performed the DRC-EB-001 study this should be clearly stated somewhere in the introduction section. 

3. Components of Lines 194 to 207 should be covered in the acknowledgment section as it provides information about the support granted for the study.

4. Can the author comment on the pharmacological outcome of increased immunogenicity and its impact on the increase or decrease of pathogenicity of EVD? Also, provide recommendations about the use of the vaccine for further development in their concluding remarks.

Comments on the Quality of English Language

NA

Author Response

Comment 1

The introduction should discuss the vaccine Ad26.ZEBOV and MVA-BN-Filo and their biochemical nature. In the current structure, it is directly focusing on clinical use. This is good. But a reader who is not a clinician and wants to know about the chemical nature, route of administration, ingredients, etc will not understand or follow the article just by the name of the vaccine.

Response:

Thank you for the comments. We are planning to publish this vaccine immunogenicity paper back-to-back with a sister paper on vaccine safety of the same trial. For background information on the Ad26.ZEBOV and MVA-BN-Filo vaccines, readers may kindly refer to the accompanying paper (reference #11) and the protocol paper (reference #3).

Comment 2

I think the authors performed the DRC-EB-001 study this should be clearly stated somewhere in the introduction section. 

Response:

Please kindly refer to the “Author Contributions” section for author responsibilities in study design, participant enrolment, study conduct, participant care , data collection and results interpretation. 

Comment 3

Components of Lines 194 to 207 should be covered in the acknowledgment section as it provides information about the support granted for the study.

Response:

This section has been removed and consolidated with the “Funding” statement that follows the “Authors Contributions” section. Other supports for the study can be found in the “Acknowledgements” section. 

Comment 4

Can the author comment on the pharmacological outcome of increased immunogenicity and its impact on the increase or decrease of pathogenicity of EVD? Also, provide recommendations about the use of the vaccine for further development in their concluding remarks.

Response:

In non-human primate models, the highest EBOV GP antibody concentrations induced by this Ad26,MVA vaccine are associated with the highest protection against death upon lethal virus challenge. Therefore, it is possible that delaying dose 2 can increase the protective effects of the vaccine but, as explained in the manuscript, since we were unable to test effectiveness, we were unable to collect any effectiveness data to explore this. Nor can we know whether this vaccine, with or without the extended interval, is protective against severe breakthrough disease.

For further use of this vaccine, we recommend that public health and immunisation services complete the two-dose schedule even if this is with an extended second dose interval, as mentioned in the “Discussion” section, because delayed administration of dose 2 of this vaccine regimen has no negative impact on immunogenicity. That said, because a person is still vulnerable to Ebola virus infection after the first dose of vaccine, we cannot recommend intentionally delaying dose 2, which would leave the recipient at risk during the interval between the two doses. We have added a statement on this to the Conclusions.

Reviewer 3 Report

Comments and Suggestions for Authors

The paper written entitled "Immunogenicity of an extended dose interval for the Ad26.ZEBOV, MVA-BN-Filo Ebola vaccine regimen in adults and children in the Democratic Republic of the Congo"  and written by Edward Man-Lik Choi and Kambale Kasonia et al., is well written and presented. It is easy to read and understand. The methodology is well presented, results are clear and conclusion is interesting and coherent to methods and results....

After well examination of conflict of interests and checking especially list of funding of this work, I recommend the publication of this interesting article and work. I just suggest for authors to:

* Add in the introduction information and references regarding the design of theses 2 vaccines ....

* Remove reference #10 (article submitted) or note instead "data not shown".

Author Response

Comment 1

After well examination of conflict of interests and checking especially list of funding of this work, I recommend the publication of this interesting article and work. I just suggest for authors to:
* Add in the introduction information and references regarding the design of theses 2 vaccines ....

Response:

Thank you for the comment. We are planning to publish this vaccine immunogenicity paper back-to-back with a sister paper on vaccine safety of the same trial. For background information on the Ad26.ZEBOV and MVA-BN-Filo vaccines, readers may kindly refer to the accompanying paper (reference #11) and the protocol paper (reference #3).

Comment 2

* Remove reference #10 (article submitted) or note instead "data not shown".

Response:

The manuscript for this reference is currently at an advanced stage of peer review by the same journal. We hope to publish the two sister papers back-to-back and provide the actual citation of this reference in the final article.